# Effect of BioPlus YC Probiotic Supplementation on Gut Microbiota, Production Performance, Carcass and Meat Quality of Pigs

**DOI:** 10.3390/ani11061581

**Published:** 2021-05-28

**Authors:** Artur Rybarczyk, Elżbieta Bogusławska-Wąs, Alicja Dłubała

**Affiliations:** 1Department of Animal Nutrition and Feed Science, Wrocław University of Environmental and Life Science, Chełmońskiego 38C, 51-630 Wrocław, Poland; 2Department of Applied Microbiology and Human Nutrition Physiology, West Pomeranian University of Technology, ul. Papieża Pawła VI 3, 71-459 Szczecin, Poland; elzbieta.boguslawska-was@zut.edu.pl (E.B.-W.); alicja.dlubala@zut.edu.pl (A.D.)

**Keywords:** pigs, *B. subtilis*, *B. licheniformis*, fattening and slaughtering traits

## Abstract

**Simple Summary:**

Use of BioPlus YC probiotic preparation containing specific Bacillus strains as a feed additive resulted in improved health and growth performance of fatteners due to a significant increase in lactic acid bacteria (LAB) count and a decrease in the number of *Enterobacteriaceae*, *Enterococcus*, *Clostridium* and *Bacillus* sp. in gut microbiota.

**Abstract:**

The objective of the study was to determine the effects of probiotic bacteria *Bacillus licheniformis* and *Bacillus subtilis* on microbiological properties of feed mixtures and on the digestive tract content as applicable to production traits and carcass characteristics of fatteners. The experiment was performed on 83,838 fatteners from four successive (insertions) productions in two groups. From the seventy eighth day of age till marketing to the slaughter plant, the pigs were supplied with BioPlus YC probiotic (Chr. Hansen) in the amount of 400 g/t. The preparation contained a complex of probiotic bacteria *Bacillus licheniformis* DSM 5749, and *Bacillus subtilis* DSM 5750 spores in a 1:1 ratio. From the fourth insertion, after reaching a body weight of approximately 112 kg, 60 fatteners were selected from each group to measure carcass quality and half of them for meat quality evaluation. Moreover, microbiological analyses in feed and colon were performed. The study showed that BioPlus YC probiotics supplementation resulted in a significantly higher count of *B. subtilis* and *B. licheniformis* in the feed, a higher count of *B. subtilis, B. licheniformis* and LAB, as well as a lower count of *Enterobacteriaceae, Enterococcus, Clostridium and Bacillus* sp. in the mucosa and in the colorectal content of the test pigs. Our work has shown that supplementation with the BioPlus YC probiotic had a positive effect on the production traits of pigs mainly by reducing mortality (2.83%, *p* = 0.010), lowering feed conversion ratio—FCR (2.59 kg/kg, *p* = 0.013), better average daily gain—ADG (0.95 kg/day, *p* = 0.002) and shorter fattening period (77.25 days, *p* = 0.019) when compared to the control group (4.19%; 2.79 kg/kg; 0.89 kg/day; 92.8 days, respectively). The addition of the specific *Bacillus* bacteria did not influence carcass and meat characteristics of the test fatteners.

## 1. Introduction

Farm animals are often exposed to environmental stresses (management methods, diet, etc.) which can cause imbalance in their intestinal ecosystem and could be a risk factor for pathogen infections. In support of specific intervention measures (e.g., animal husbandry, hygienic practices, feeding and transport prior slaughter), feeding probiotic supplements could be used in an integrated approach to improve the food safety from the initial step of the “farm to fork” food chain, which means the maintenance of a healthy intestinal ecosystem [1]. This, and the growing consumer demand for more naturally produced and high-quality meat, has encouraged producers to use natural and non-chemical forage supplements, which positively influence animal health, productivity and meat quality [2,3]. Therefore, and as an alternative to more commercial pig production, supplementing probiotics have gained interest in recent years.

Hong et al. [4] reported that various *Bacillus* spp. could be used as antibiotic alternatives for humans and animals. *Bacillus* are Gram-positive, spores-forming microorganisms that produce antibiotics, bacteriocins and lytic enzymes with antimicrobial activity; they also secrete amylolytic and pectinolytic enzymes that support digestive functions of the gut, as well as produce essential amino acids and vitamins [5]. Isolates sporulated efficiently in the laboratory, and the resulting spores were tolerant to simulated gastrointestinal tract conditions. They also exhibited antimicrobial activity against a broad spectrum of bacteria, including food spoilage and pathogenic organisms. They cannot colonize in the gastrointestinal tract, but they can stimulate the growth of Lactobacilli through production of catalase and subtilisin [6]. Several studies suggested that dietary addition of *Bacillus* spp. could lead to increased growth performance and improved health of pigs [7,8,9,10,11,12], nutrient digestibility [13], humoral and cell-mediated immune responses [14] and improved the gastrointestinal microbiota [15]. However, little is known about the effect of pig supplementation using *Bacillus* spp. probiotics on the carcass and meat quality. The up to date and a few studies indicate the beneficial effect of pig supplementation with specific probiotic strains of *Bacillus* spp. on the slaughter value [9,12] and meat quality [12,16].

The aim of this study was to determine the effects of probiotic bacteria *Bacillus licheniformis* and *Bacillus subtilis* on gut microbiota, fattening results, carcass and meat quality of pigs.

## 2. Materials and Methods

### 2.1. Animals

The research was carried out on fatteners derived from the offspring of Landrace–Yorkshire (LY) sows and the purebred Duroc boars coming from pig production farm in the Pomeranian Voivodship (Poland). All experimental pigs were housed under the same environmental conditions in a non-bedding system. During the fattening period, pigs were kept under the same environmental conditions and were fed the same balanced dry loose complete feed mixture and fed *ad libitum*. The list of components and feedstuff chemical composition are presented in Table 1. The energy value of the cereals was calculated based on digestibility coefficients listed in the Nutrient Requirements for Swine [17].

The experiment was conducted in two similar piggeries with different total volumes. Each building was equipped with a gravity ventilation system and housed one analyzed group of fatteners. In both piggeries, washing and disinfection (Biogel and Virocid) occurred after each sale, before re-insertion of fatteners. Moreover, during the fattening period, from the 1st to the 12th week, the ambient temperature gradually decreased from 19 °C to 16 °C. The pen area for a single head was 0.65 m^2^, and there was an attempt to keep the sex ratio at 1:1. The control group fattening house had a production area of 6052 m^2^ and was allocated to 354 pens (approx. 14 m^2^ and 20.5 m^2^, total number of fatteners 9311). However, the building of the probiotic group had a production area of 8383.22 m^2^ and was allocated to 688 pens (approx. 10 m^2^ and 14 m^2^, total number of fatteners 12,897).

The experiment was performed on 83,838 fatteners from four successive (insertions) production of each group (pig placement was done every 3 months), equally divided into Control (n = 33,264) and BioPlus YC (n = 50,574) group. From the seventy eighth day of age till marketing to the slaughter plant, the pigs were supplied with BioPlus YC probiotic (Chr. Hansen) in the amount of 400 g/t. The preparation contained a complex of probiotic bacteria *Bacillus licheniformis* DSM 5749—1.6 × 10^9^ CFU/g, and *Bacillus subtilis* DSM 5750—1.6 × 10^9^ CFU/g; spores in a 1:1 ratio. Neither control group nor pigs supplemented with the BioPlus YC probiotic were treated with antibiotics for therapeutic purposes. The fattening efficiency results for four insertions were entered into Cloudfarms.

From the fourth insertion (respectively: Control: n = 8248 and BioPlus YC: n = 10,728), 60 fatteners were selected from each group (15 pens × 4 fatteners) according to similar body weights and sex balanced (barrows and gilts), to slaughtering performance, defined as carcass quality (total number 120 carcasses). Microbiome was evaluated on 10 samples from every group.

### 2.2. Performance Traits

Based on the report generated from Cloudfarms (a cloud-based Pig Production Management System), body weight of fatteners at the beginning (BW1) and at the end of fattening period (BW2), fattening time, average daily gain (ADG), average daily feed intake (ADFI), feed conversion rate (FCR) and mortality were determined. Dressing yield was determined on the basis of BW2 and hot carcass weight (HCW) obtained from the Meat Processing Plants.

### 2.3. Carcass and Meat Quality

After reaching a body weight of approximately 112 kg, from fatteners from an insertion no. IV selected 30 barrows and 30 gilts (each group) to measure carcass quality and half of them for meat quality evaluation. The duration of fasting time before slaughter was 24 h. The average temperature during the transport of fatteners to Meat Processing Plants was 22.8 °C. After unloading, the pigs were given 16 h of pre-slaughter rest at an ambient temperature of 15–16 °C.

On the slaughter line, after the pigs were stunned (Butina CO_2_ gas stunning system, Marel, Iceland), the lean meat percentage in the carcass, the ham, loin, shoulder and belly, were non-invasively ultrasonically measured (AutoFom, SFK Technology, Herlev, Denmark). Before cooling, the hot carcass weight was determined with an accuracy of 100 g. Next, the carcasses were progressively chilled for 24 h. At first, they were cooled at a temperature of +1 °C for 7–8 h, and then after filling the chilling room, the carcasses were cooled at a temperature between −3 °C and −4 °C for 6–7 h, and then at 4–6 °C for the remaining time (about 10 h). In the cooling chamber, 30 carcasses of similar weight (HCW: 90 ± 5 kg) were selected from the probiotic and 30 from the control group to determine meat quality traits.

After cooling the carcasses, as the right halves were fabricated into the basic elements, *longissimus lumborum* (LL) muscle samples were collected from the 1st–4th lumbar vertebral regions of each half-carcass for testing. The LL muscle samples, packaged in labeled foil pouches, were transported in thermoses to the laboratory and stored in a refrigerator at 4 °C. Subsequently, 3 slices each 3 cm thick (100 g) were cut out of the muscle starting from the cranial end, to determine the drip loss, pH and color traits. The remainder of the muscle was packed into pre-labeled plastic bags and frozen at −19 °C for approximately one month to determine shear force.

The following qualitative determinations were made on fresh meat:The pH was measured at 35 min and 48 h *post mortem* (p.m.) in LL muscle and pH at 24 h in *semimemembranosus* muscle (SM), using a portable pH-meter equipped with a temperature sensor (CP-411 pH-meter, Elmetron, Zabrze, Poland). pH 35 min and pH 24 h p.m. were measured in a cold room, on right half-carcasses.Electrical conductivity in LL muscle was determined at 24 h (EC_24_) in a cold room on the right halves using the LF-Star device (Ingenieurbüro Matthäus, Hamburg, Germany).Drip loss was determined by the method of Prange et al. [18]. 24 h, LL muscle samples weighing 50 g (cut out from the middle part of the 3 cm thick slices) were put into plastic bags and stored at 4 °C. Drip loss was defined as % loss in mass after 1 day (48 h) of storage.The measurement of color was performed on freshly cut LL muscle slices at 48 h, after 20 min blooming period at 4 °C. Color lightness (L*), redness (a*), yellowness (b*), chroma (C*) were determined by a HunterLab Mini Scan XE Plus 45/0 (HunterLab Inc., Reston, VA, USA), equipped with a standard illuminant D65 and 10° Standard Observer.

### 2.4. Shear Force

A fragment of LL muscle (about 300 g) was removed from the freezer, thawed at 4 °C for about 24 h. Each sample was heated in a water bath at 80–81 °C until reaching an internal temperature of 72 °C; and subsequently was cooled to 20 °C. Shear force was measured using a Warner-Bratzler apparatus (WB) manufactured at the Baking Industry Research Centre (Bydgoszcz, Poland). Cylinder-shaped meat samples cut out with a cork borer with a diameter of 1.0 cm (along the muscle fibers) were placed in a triangular recess under five blades of the tenderness measuring instrument, which then recorded the maximum force (expressed in kilograms) required for cutting through the meat. The final result for each sample was the average of three consecutive trials.

### 2.5. Proximate Analysis

The basic chemical composition of LL muscles was determined in accordance with the official analytical methods of the AOAC [19]: moisture content by the oven-drying of 2 g samples at 102 °C to a constant weight; crude protein content by the classical macro-Kjeldahl method and intramuscular fat content by petroleum ether extraction using a Soxhlet apparatus. The total mineral (ash) content was determined by incineration at 550 °C.

### 2.6. Microbiological Determinations

All microbiological analyses of the samples (feed, colon mucous membranes, colon content) were performed in accordance with accepted standards. The following ISO standards were used to determine specific groups of microorganisms: total bacterial count (TBC)—[20]; total yeast and mold count (TYMC)—[21]; total count of *Enterobacteriaceae* (TCE), capable of degrading trichloroethylene—[22]; *Staphylococcus* sp. (STP)—[23]; *Salmonella* sp. [24]; *Listeria monocytogenes*—ISO [25], *Enterococcus* sp. (TCC)—[26]; lactic acid bacteria (LAB)—[27]; anaerobic spore-forming bacteria—*Clostridium* (CL)—[28]; aerobic spore-forming bacteria—*Bacillus* sp.—[29], using Mannitol Egg Yolk Polymyxin Agar (0.1% meat extract, 1.0% peptone, 1.0% mannitol, 1.0% sodium chloride, 0.0025% phenol red, polymyxin B; MYP, OXOID). In order to ensure the reliability of microbiological tests in relation to particular standards, [30] was adhered to.

Samples of Rosta and Finisher feed were taken for microbiological tests immediately after preparation (mixing of ingredients) for the pigs in the control and supplemented with BioPlus YC probiotic. Two samples were taken from Rosta and Finisher feed (an interval of two weeks), and the obtained results were averaged. Moreover, the microbiological analysis was performed on pure BioPlus YC probiotic. For microbiological determinations, 10 cm sections of proximal colon were taken on the slaughter line during evisceration from 10 randomly selected pigs from probiotic and control group, with an equal sex ratio [31].

In the case of determination of pathogenic and potentially pathogenic bacteria (*Salmonella* sp., *Staphylococcus* sp., *L. monocytogenes*), qualitative determinations were performed, both in the analyzed feed and in the collected sections of the digestive tracts (on the same number of sampling).

Identification of *Bacillus* sp.: Bacterial strains isolated on the MYP medium were subjected to a diagnostic analysis considering biochemical and phenotypic features [32]. Species affiliation of all isolated strains, initially classified to *Bacillus subtilis* and *Bacillus licheniformis*, was confirmed with the PCR technique. Cultures were incubated on the TSB medium (1.7% tryptone, 0.3% soy peptone, 0.5% sodium chloride, 0.25% dipotasium phosphate; Scharlau) at 30 °C for 24 h. Genomic DNA was isolated following the protocol of the Genomic Mini AX Bacteria (A&A Biotechnology) using mutanolisine (Sigma-Aldrich). Extracted DNA was amplified using universal primers OPR-13 (5′-GGACGACAAG-3′) for *B. licheniformis* [33] and A-19 (5′-AGTCAGCCAC-3′) for *B. subtilis* [34]. The PCR reaction was conducted in 25 µL of the reaction mixture containing: 10.0 µL of MIX PCR (A&A Biotechnology), 1.0 µL of each primer, and 2.0 µL of DNA template. The PCR reaction was conducted in a thermocycler (Eppendorf) under the following thermal profile: initial denaturation at 95 °C/15 min, annealing 63 °C/45 s, extension 72 °C/2 min, for a total of 30 cycles [35]. Amplification products were separated electrophoretically in 2.0% agarose gel (Prona Agarose Plus) with ethidium bromide (0.5 µL/mL) (Bio-Rad, Hercules, CA, USA). Results of the electrophoretic separation were visualized in UV rays in a GelDoc apparatus (Bio-Rad, Hercules, CA, USA).

### 2.7. Statistical Analysis

The obtained data for carcass and meat quality was analyzed statistically by means of a Statistica 13.1 PL software using a one-factor analysis of variance. Microbial results were given as the total number of microorganisms, expressed in log colony forming units. A detailed comparison of means was performed using the Tukey’s test at *p* ≤ 0.01 and *p* ≤ 0.05. The tables show average values and their standard errors.

## 3. Results

### 3.1. Production Performance

Analysis of fattening results generated from the Cloudfarms-herd management system (Table 2) revealed significant lower mortality (2.83 vs. 4.19%, *p* ≤ 0.05) and higher ADG (0.95 vs. 0.89%, *p* ≤ 0.05), lower FCR (2.59 vs. 2.79%, *p* ≤ 0.05), and a shorter fattening time (77.25 vs. 92.80 days, *p* ≤ 0.05) in pigs supplemented with the BioPlus YC probiotic compared to fatteners from the control group.

### 3.2. Microbiological Tests of the Feeds and the Digestive Tract

The results of microbiological analyses of feed samples and colon sections from the pigs did not indicate any presence of pathogenic or potentially pathogenic microorganisms (*Salmonella* sp., *Staphylococcus* sp. and *L. monocytogenes*) noted in the microbiological safety criteria [36,37].

The analysis of microbiological results showed significant differences in the numbers of *B. subtilis* and *B. licheniformis*, which were higher in the feed supplemented with BioPlus YC probiotics than in the control group feed (Table 3). No significant differences were observed between the analyzed feeds in total bacterial count (TBC) and total yeast and mold count (TYMC) and lactic acid bacteria (LAB).

Microbiological analyses of samples taken from the intestinal mucosa and digestive tract (Table 4) showed no significant differences in the number of TYMC and *Staphylococcus* (STP) between the control group and the group supplemented BioPlus YC probiotics. There was a significant increase *B. subtilis*, *B. licheniformis* and LAB and, at the same time, a significant decrease TBC, including Enterobacteria (TCE), *Enterococcus* (TCC), *Clostridium* (CL) and *Bacillus* sp. in mucosal colonization and gastrointestinal contents.

### 3.3. Carcass and Meat Quality

Based on the slaughter carcass value (Table 5), there were no significant differences between the group supplemented with the BioPlus YC probiotic vs. the control group in terms of leanness (meatiness) of the individual cuts such as ham, loin, shoulder and belly. Moreover, no significant differences were found both the groups in terms of the basic chemical composition, physicochemical properties and shear force of LL muscle (Table 6).

## 4. Discussion

In the healthy animal, a balance of micro-organisms in the gastrointestinal tract helps in efficient digestion and maximum absorption of nutrients, and increases the body’s resistance to infectious diseases [38,39]. The presented study showed that BioPlus YC probiotics supplementation resulted in a higher count of *B. subtilis*, *B. licheniformis* and LAB as well as a lower count of *Enterobacteriaceae*, *Enterococcus*, *Clostridium* and *Bacillus* sp. in the mucosa and colorectal content. The obtained results can be explained by the fact that *Bacillus* spp. is not a principal member of the normal physiological intestinal flora and could not colonize the intestine for long periods, it consumes oxygen rapidly and reduces pH, which favors *Lactobacillus* and inhibits *E. coli* and *Salmonella* [40]. In other studies, application of *Bacillus* (BioPlus 2B) probiotic treatments resulted in reduced numbers of fecal *Salmonella* and *E. coli* as well as increased *Lactobacillus* spp. and *Bacillus* spp. counts compared to control [14]. Additionally, Sheng et al. [41] found that groups of fattening pigs supplemented with *B. subtilis* natto and *B. coagulans* had a higher concentration of Lactobacilli, and a lower concentration of *E. coli* and *Clostridium* in the feces from the rectum.

In the present study, analysis of production results revealed lower mortality and FCR, higher ADG and shorter fattening period in pigs supplemented with the BioPlus YC probiotic compared to fatteners from the control group. Studies carried out by Alexopoulos et al. [8] showed that BioPlus 2B feed additive improved gilts or sows reproductive performance. Moreover, certain blood and milk parameters were improved, which resulted in a positive effect in piglets’ health and performance. Other research by Alexopoulos et al. [9] proved that BioPlus 2B supplementation of pigs had a positive impact on a lower morbidity and mortality of weaned piglets and improved the fattening performance-ADG by up to 8%, and feed use efficiency by up to 10% in grower and finisher pigs in a dose-dependent manner. Tests performed by Davis et al. [11] with the use of MicroSource^®^ probiotic (*B. subtilis* and *B. licheniformis*) for pig’s feeding, revealed an improved feed efficiency and decreased the time required to disperse a swine manure mat sample. Furthermore, the above probiotic improved gain and decreased mortality of pigs during the growing–finishing period. Balasubramanian et al. [12] reported that the addition of commercially available *Bacillus*-based probiotic (SynerZymeH10), containing *B. coagulance*, *B. lichenformis* and *B. subtilis*, prepared at 0.2%, for the pig’s feed, was effective in improving the growth performance-ADG and feed intake/gain (F/G), nutrient digestibility of dry matter. Additionally, Chen et al. [10] found that dietary supplementation with Bacillus-based probiotic (*B. subtilis* and *B. coagulans*) preparation at the level of 0.2% is effective in improving the growth performance and reducing fecal NH_3_-N and butyric acid concentrations in finishing pigs. Cui et al. [41] concluded that the addition of *B. subtilis* improves the growth performance, which is evidenced by improved ADG and ADFI and decreased F/G. Moreover, in the studies Ahmed et al. [14], application of *Bacillus*-based probiotics (BioPlus 2B) resulted in improvement of FCR, but had no significant effect on ADG and ADFI of weaned piglets. On the contrary, Wang et al. [42] reported that supplementation with BioPlus 2B can reduce the slurry NH_3_ emission, but not H_2_S and mercaptan emission in growing pigs without impacting the growth performance.

In the present study, there are no differences between the group supplemented with the BioPlus YC probiotic and the control in terms of the slaughter carcass value of pigs. In other studies, it was shown that BioPlus 2B probiotic improves the carcass quality of fatteners [9]. Additionally, Balasubramanian et al. [12] discovered in their study that the supplementation of *Bacillus* spp. probiotic improved the carcass weight and carcass grade, but not the backfat thickness. However, Cui et al. [42] reported, that pigs supplemented with probiotic containing *B. subtilis*, higher average backfat and *longissimus* muscle area as compared to values in the control group, at similar body weights before the slaughter.

In present research, no significant differences were found between the group of fattening pigs supplemented with the BioPlus YC probiotic and the control group in terms of the basic chemical composition, physicochemical properties and shear force of LL muscle. The slight of influence of probiotic bacteria *B. subtilis*, *B. coagulance* and *B. licheniformis* on meat quality is confirmed by Cho et al. [16] and Balasubramanian et al. [12]. In both studies, the authors showed the beneficial effect of *Bacillus* spp. on the color of meat in visual assessment and its higher redness (a*). However, Sheng et al. [40] reported that the supplementation of pig feed with *B. subtilis* natto significantly improves meat quality (pH 24 h p.m.), increases antioxidant function and reduces skatole production, while its combination with *B. coagulans* enhanced these effects.

## 5. Conclusions

Our results indicate that the administration of a BioPlus YC probiotic preparation containing specific *Bacillus* strains to pigs resulted in a higher health status and fattening performance. The *B. subtilis* and *B. licheniformis* dosage had a significant effect on the gut microbiota through a significant increase in LAB count and a decrease in the number of *Enterobacteriaceae, Enterococcus, Clostridium* and *Bacillus* sp. This change in microbiota positively influenced the production results of fattening pigs, mainly by reducing mortality, improving FCR and ADG. We observed no benefits of our treatments for carcass and meat quality parameters. The obtained results indicate that the use of the BioPlus YC probiotic was efficacious in improving pig production economics.

## Figures and Tables

**Table 1 animals-11-01581-t001:** Composition of the experimental diets.

Items	Rosta 20–50 kg Body Weight	Finisher 45–100 kg Body Weight
Ingredient (g/kg on a DM basis)		
Wheat grain	116.0	116.0
Barley grain	106.0	106.0
Triticale grain	106.0	106.0
Wheat bran	-	156.0
NaCl	7.0	6.1
Complementary feed	12.5	10.0
Other ^1^	652.5	499.9
Chemical composition		
Metabolizable energy (MJ/kg)	11.80	11.60
Net energy (MJ/kg)	9.74	9.51
Crude protein (g/kg)	174.2	158.8
Total fibre (g/kg)	38.0	46.6
Crude fat (g/kg)	33.8	30.3
Calcium (g/kg)	7.7	5.8
Total phosphorous (g/kg)	6.0	4.5
Lysine (g/kg)	12.6	9.8
Methionine (g/kg)	3.9	2.8
Methionine + cysteine (g/kg)	7.5	6.6
Threonine (g/kg)	8.0	6.4
Tryptophan (g/kg)	2.4	1.9
Isoleucine (g/kg)	6.6	5.8
Valine (g/kg)	7.9	7.2

^1^ Other: post-extraction soy meal, toasted, post-extraction rapeseed meal, rapeseed rape EP-100, narrow-leaved lupine, animal fat, fine grained chalk (CaCO_3_ min. 94%. Ca—37.6%), phosphate 1-CA2 (additive contains min. 22% P and 15% Ca)—protected feed formulation.

**Table 2 animals-11-01581-t002:** Results of fattening and slaughtering performance from Pig Production Management System (Cloudfarms).

Parameter	Control	BioPlus YC	*p*-Value
BW1 (kg)	28.05 ± 1.01	33.15 ± 2.36	0.094
BW2 (kg)	110.23 ± 0.68	112.00 ± 2.29	0.486
Mortality (%)	4.19 ^a^ ± 0.28	2.83 ^b^ ± 0.68	0.010
Fattening period (days)	92.80 ^a^ ± 2.03	77.25 ^b^ ± 4.41	0.019
ADG (kg/day)	0.89 ^B^ ± 0.01	1.02 ^A^ ± 0.03	0.002
FCR (kg/kg)	2.79 ^a^ ± 0.03	2.59 ^b^ ± 0.05	0.013
ADFI (kg/day)	2.47 ± 0.04	2.48 ± 0.08	1.000
consumption of feed per produced fattener (kg)	229.39 ^a^ ± 5.20	191.23 ^b^ ± 13.42	0.038
HCW (kg)	83.90 ± 1.63	86.00 ± 1.81	0.421
Dressing yield (%)	76,12 ± 1.50	76.79 ± 0.12	0.676

Mean values in rows marked by different letters differ significantly: ^A,B^: *p* ≤ 0.01; ^a,b^: *p* ≤ 0.05. BW1: body weight of fatteners at the beginning; BW2: body weight of fatteners at the end of fattening period; ADG: average daily gain; ADFI: average daily feed intake; FCR: feed conversion rate; HCW: hot carcass weight. The number of the insertions of fatteners in each group—4.

**Table 3 animals-11-01581-t003:** Composition of microbiological fractions in feed.

Microbiological Fractions (log_10_/g)	Control	Mixture BioPlus YC and Feed	BioPlus YC (Preparation)	*p*-Value
TBC	4.9 ^a^ ± 0.37	5.4 ^a^ ± 0.58	1.1 ^b^ ± 0.17	0.029
TYMC	1.3 ± 0.17	1.6 ± 0.59	1.3 ± 0.69	0.281
LAB	5.4 ^a^ ± 0.28	3.4 ^a^ ± 0.48	1.9 ^b^ ± 0.29	0.049
*Bacillus* sp.	3.5 ^a^ ± 0.39	2.5 ^a^ ± 0.40	<2.0 ^b^	0.011
*B. subtilis*	<2.0 ^c^	5.1 ^b^ ± 0.58	8.9 ^a^ ± 0.39	0.028
*B. licheniformis*	<2.0 ^c^	4.89 ^b^ ± 0.27	9.6 ^a^ ± 0.36	0.021

^a,b,c^ Mean values in rows marked by different letters differ significantly at *p* ≤ 0.05. TBC: total bacterial count; TYMC: total yeast and mold count; LAB: lactic acid bacteria. The number of sample in each group—4.

**Table 4 animals-11-01581-t004:** Colon microbiota composition.

**Microbiological Fractions (log_10_/g)**	**Control**	**BioPlus YC**	***p*-Value**
Proximal colon mucosa			
TBC	7.62 ^a^ ± 0.57	6.38 ^b^ ± 0.79	0.026
TYMC	2.57 ± 0.96	2.25 ± 1.05	0.574
LAB	4.00 ^b^ ± 0.80	5.70 ^a^ ± 0.81	0.039
TCE	9.58 ^a^ ± 0.85	7.48 ^b^ ± 1.81	0.042
TCC	6.47 ^a^ ± 1.52	3.18 ^b^ ± 0.72	0.011
STP	4.63 ± 0.51	4.18 ± 0.37	0.115
CL	5.95 ^a^ ± 1.33	4.18 ^b^ ± 1.41	0.011
*Bacillus* sp.	3.3 ^a^ ± 0.64	1.02 ^b^ ± 1.52	0.011
*B. subtilis*	<2.0 ^b^	4.03 ^a^ ± 0.61	0.000
*B. licheniformis*	<2.0 ^b^	3.74 ^a^ ± 0.72	0.000
Digestive tract of proximal colon			
TBC	7.79 ^a^ ± 0.64	6.54 ^b^ ± 0.81	0.000
TYMC	2.82 ± 1.13	2.36 ± 1.15	0.349
LAB	4.03 ^b^ ± 0.94	5.89 ^a^ ± 0.76	0.000
TCE	9.99 ^a^ ± 0.37	7.73 ^b^ ± 1.81	0.000
TCC	7.05 ^a^ ± 1.21	3.35 ^b^ ± 0.62	0.028
STP	4.93 ± 0.56	4.21 ± 0.43	0.359
CL	6.37 ^a^ ± 1.16	4.48 ^b^ ± 0.94	0.000
*Bacillus* sp.	4.16 ^a^ ± 0.74	2.15 ^b^ ± 1.54	0.005
*B. subtilis*	<2.0 ^b^	2.95 ^a^ ± 1.23	0.002
*B. licheniformis*	<2.0 ^b^	2.29 ^a^ ± 0.72	0.018

^a,b^ Mean values in rows marked by different letters differ significantly at *p* ≤ 0.05. TBC: total bacterial count; TYMC: total yeast and cold mount; LAB: lactic acid bacteria; TCE: total count of *Enterobacteriaceae*; TCC: total count of *Enterococcus*; STP: total count of *Staphylococcus*; CL: total count of *Clostridium*. The number of sample in each group—10.

**Table 5 animals-11-01581-t005:** Slaughter carcass value.

Traits	Control	BioPlus YC	*p*-Value
HCW (kg)	89.05 ± 0.43	89.06 ± 0.38	0.990
Lean meat in carcass (%)	56.65 ± 0.32	56.58 ± 0.32	0.879
Lean meat in ham (%)	59.54 ± 0.33	59.87 ± 0.30	0.467
Lean meat in loin (%)	53.06 ± 0.46	53.45 ± 0.41	0.535
Lean meat in shoulder (%)	57.28 ± 0.26	57.38 ± 0.23	0.777
Lean meat in belly (%)	52.15 ± 0.40	52.76 ± 0.33	0.241

HCW: hot carcass weight. The number of sample in each group—60.

**Table 6 animals-11-01581-t006:** Meat quality and basic chemical composition.

Traits	Control	BioPlus YC	*p*-Value
pH_35 min_	6.58 ± 0.02	6.56 ± 0.02	0.564
pH_24_ SM	5.98 ± 0.05	5.92 ± 0.03	0.232
pH_48_	5.72 ± 0.04	5.65 ± 0.02	0.074
Drip loss (%)	2.49 ± 0.18	2.69 ± 0.17	0.602
EC_24_ (mS/cm)	3.94 ± 0.22	4.22 ± 0.19	0.463
L*	56.96 ± 0.54	57.62 ± 0.44	0.206
a*	5.47 ± 0.11	5.35 ± 0.13	0.553
b*	13.86 ± 0.15	14.14 ± 0.14	0.187
C*	14.94 ± 0.10	15.12 ± 0.15	0.229
Shear force, kg	4.09 ± 0.13	4.06 ± 0.07	0.906
Dry matter (%)	25.53 ± 0.16	25.29 ± 0.12	0.216
Total protein (%)	20.96 ± 0.09	20.76 ± 0.10	0.158
Intramuscular fat (%)	2.57 ± 0.13	2.49 ± 0.12	0.678
Ash (%)	1.18 ± 0.02	1.21 ± 0.02	0.158

L*—lightness; a*—redness; b*—yellowness; C*—saturation; EC—electrical conductivity. The number of sample in each group—30.

## Data Availability

The data presented in this study are available on request from the corresponding author. The data are not publicly available due to privacy.

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
