# Peer review of "Effect of BioPlus YC Probiotic Supplementation on Gut Microbiota, Production Performance, Carcass and Meat Quality of Pigs"

_animals, 2021, doi:10.3390/ani11061581_

Round 1

Reviewer 1 Report

All my previous comments have been corrected, or clarified.

In the tables, the total value column has been replaced by P-Value. In my opinion, this is double reporting the P Value size in one table, which is unnecessary. In my opinion the letter signs should remain next to the mean values, to which the p-value is assigned (described under the table).

Author Response

In the tables, the total value column has been replaced by P-Value. In my opinion, this is double reporting the P Value size in one table, which is unnecessary. In my opinion the letter signs should remain next to the mean values, to which the p-value is assigned (described under the table).: We understand the reviewer's arguments to remove P-value from tables. However, other reviewers pointed to the need to include them in the table. We suggest leaving the P-value in the tables.

Reviewer 2 Report

The manuscript has evolved considerably in terms of characterizing its content, especially in terms of material and methods and results. Introduction, discussion and conclusions sections that usually reflect the state of the art of the topic and the scientific argumentation of the answers obtained regarding the investigated problem continued at the same level as before. The description of the treatment name using the commercial name of the tested product does not seem adequate to me.

Author Response

Introduction, discussion and conclusions sections that usually reflect the state of the art of the topic and the scientific argumentation of the answers obtained regarding the investigated problem continued at the same level as before.: We cannot agree with the reviewer's argument that the parts of the thesis on scientific argumentation have not been corrected. In accordance with the comments and recommendations of the reviewer, we have made the corrections mentioned below. In the Introduction section, as no antibiotics and LAB probiotics were used, we changed the text from “For several decades, antibiotics and chemotherapeutics in prophylactic dosages have been used in animals. However, there have been growing concerns about the risk of developing cross-resistance and multiple antibiotic resistance in pathogenic bacteria in both humans and livestock [1]. This, and the growing consumer demand for safe and high-quality meat, has encouraged producers to use natural and safe non-chemical forage supplements which positively influence animal health, productivity and meat quality [2]. As an alternative, supplementing probiotics gained interest in recent years. Lactobacillus being the most commonly used probiotic agent, improves the growth performance, feed conversion efficiency, nutrient utilization, intestinal microbiota, gut health and regulates immune system in pigs [3]” on „Farm animals are often exposed to environmental stresses (management methods, diet, etc.) which can cause imbalance in their intestinal ecosystem and could be a risk factor for pathogen infections. In support to specific intervention measures (e.g. animal husbandry, hygienic practices, feeding and transport prior slaughter), feeding probiotic supplements could be used in an integrated approach to improve the food safety from the initial step of the “farm to fork” food chain, which means the maintenance of a healthy intestinal ecosystem [1]. This, and the growing consumer demand for more naturally produced and high-quality meat, has encouraged producers to use natural and non-chemical forage supplements, which positively influence animal health, productivity and meat quality [2,3]. Therefore, and as an alternative to more commercial pig production, supplementing probiotics gained interest in recent years”.

We also made changes to Conclusions at the reviewer's request, was in the previous version of the publication „Our results indicate that the administration of a BioPlus YC probiotic preparation containing specific Bacillus strains, resulted in a higher health status and fattening performance. The B. subtilis and B. licheniformis dosage had a significant effect on the gut mi-crobiota through a significant increase in LAB count and a decrease in the number of Enterobacteriaceae, Enterococcus, Clostridium and Bacillus sp which positively influenced on the production results of fattening pigs, mainly by limiting the mortality, lower FCR, higher ADG and shorter fattening period but not affected on carcass and meat quality. The ob-tained results indicate that the use of the BioPlus YC probiotic is associated with higher economic efficiency - faster fattening, lower feed consumption by pigs”. It is now „Our results indicate that the administration of a BioPlus YC probiotic preparation containing specific Bacillus strains to pigs, resulted in a higher health status and fattening performance. The B. subtilis and B. licheniformis dosage had a significant effect on the gut microbiota through a significant increase in LAB count and a decrease in the number of Enterobacteriaceae, Enterococcus, Clostridium and Bacillus sp. This change in microbiota posi-tively influenced on production results of fattening pigs, mainly by reducing mortality, improving FCR and ADG. We observed no benefits of our treatments for carcass and meat quality parameters. The obtained results indicate that the use of the BioPlus YC probiotic was efficacious in improving pig production economics”. In the Discussion section, we also edited and completed the text, especially the last paragraph.

The description of the treatment name using the commercial name of the tested product does not seem adequate to me.: In accordance with the reviewer's remark, the title of the publication has been changed so that it does not appear in the name of the commercial product. Moreover, the commercial name of the Bacillus strains was removed in the sentence concerning the purpose of the publication as well as Abstract.

Reviewer 3 Report

The Authors greatly improved the manuscript and answered satisfactorily to the previous version remarks. Thus, in my opinion it is now suitable for publication. Just a small typo: at line 92 I think that the subject of the sentence is missing ("selected from each group").

Author Response

The Authors greatly improved the manuscript and answered satisfactorily to the previous version remarks. Thus, in my opinion it is now suitable for publication. Just a small typo: at line 92 I think that the subject of the sentence is missing ("selected from each group").: In accordance with the reviewer's comment, the sentence has been corrected.

Reviewer 4 Report

This manuscript by Rybarczyk and colleagues have determined the effect of BioPlus YC probiotic on gut microbiota, production performance, carcass and meat quality of pigs. The works show that supplementation with the BioPlus YC probiotic has a positive effect on the production traits of pigs. The following changes could improve the quality of the paper.

  1. Please specify the “BioPlus YC” from the scientific point in the title.
  2. Abstract, please describe the details the experimental design.
  3. Abstract, Please add the P values (P = 0.05?) for the significant changes.
  4. Please remove the all the “DSM” from the main text, only left it in the material is fine. Do not make it looks too commercial.
  5. In line 115, please correct “of + 1oC…”
  6. Please describe the how many replicates for each treatment in the experimental design section? It is hard to understand.
  7. Please specify the replicates (n=?) in each table notes, but not in the table.
  8. Some of the table missed the table note. Please check them.
  9. Please write all the “P” as italic, and the space need to be added before and after “<”.

Author Response

  1. Please specify the “BioPlus YC” from the scientific point in the title.: In accordance with the reviewer's remark, the title of the publication has been changed so that it does not appear in the name of the commercial product.
  2. Abstract, please describe the details the experimental design.: OK, research methodology was added to Abstract.
  3. Abstract, Please add the P values (P = 0.05?) for the significant changes.: OK, corrected.
  4. Please remove the all the “DSM” from the main text, only left it in the material is fine. Do not make it looks too commercial.: The commercial name of the Bacillus strains was removed in the sentence concerning the purpose of the publication as well as Abstract.
  5. In line 115, please correct “of + 1oC…”: OK, corrected.
  6. Please describe the how many replicates for each treatment in the experimental design section? It is hard to understand.: In each treatment were 4 replicates. The sentence has been corrected.
  7. Please specify the replicates (n=?) in each table notes, but not in the table.: OK, corrected.
  8. Some of the table missed the table note. Please check them. OK, an explanation of abbreviations for the color and electrical conductivity characteristics is added below table 6.
  9. lease write all the “P” as italic, and the space need to be added before and after “<”. OK, corrected.

Round 2

Reviewer 4 Report

No further comments.

This manuscript is a resubmission of an earlier submission. The following is a list of the peer review reports and author responses from that submission.

Round 1

Reviewer 1 Report

Interesting paper, broadening the knowledge on the influence of probiotic supplementation to pig feed on intestinal microflora and fattening results, carcass and meat quality.

The work needs a few corrections and clarifications.

line 14 and 21. - please elaborate on the acronym "LAB"

line 55 - The authors cite literature item no. 8, which should be 9 (??)

line 83  - it says "Dressing percentage" , it should be "Dressing yield", also in the table 2 the name of this traits should be corrected

line 102 - please clarify whether the whole LL muscles were taken or in a specific length or weight

line 104 - please specify here the weight of the sample with 3 cm width.

line 109 and 112 - pH was measured at 35 minutes ? not  at 45 minutes?

line 183 - please specify at which levels of significance the Turkey test was performed,

table 1.  - please explain what the columns "title 1", "title 2" , "title 3" mean, I suggest removing them.

line 217 - In the legend under Table 1, the * should be replaced by 1.

table 2 and 3 - please, in the legend below the tables, describe the abbreviations used in the tables.

table 6 - the values of parameter a and b in the "Total" column are entered incorrectly.

line 275 - please correct the citation record from "(2016)" to [12].

Please provide information about the ethical committee's approval to perform the study.

Reviewer 2 Report

There are a lot of concerns surrounding this manuscript.

Abstract:

L20: All species names must be in italics. It is also observed in other moments of the manuscript (L71)

L22 - 26: “Studies have shown that ..” The way it was written suggests a literature review of other studies, which does not fit in the abstract.

Keywords:

L28: do not use words present in the title

Introduction

It is a biased essay, whose construction of the problem is not directly related.

This section discusses the use of antibiotics and chemotherapeutics and brings the use of probiotics as an alternative, however, this has not been investigated between treatments. There is reference to Lactobacillus, which has also not been investigated.

Materials and Methods

There is no information regarding the experimental design, number of repetitions, animals per experimental unit, area per animal housed, ambience, experiment location, ethics committee ...

What did you consider as an experimental unit for performance, Carcass and meat quality, Shear force, Proximate analysis, and Microbiological determinations?

The treatments are presented very superficially and are not clear. There is a key point: were there antimicrobials in the control group's diet? Is it correct to characterize it as negative control or positive control? If it is a diet without antimicrobials, the entire introduction section should be rewritten because from the beginning it suggests to the reader that it is an evaluation of probiotics against antibiotics.

The level of health challenge is a relevant factor when assessing the use of probiotics, mainly regarding the microbiological characterization, however, in this manuscript there was no characterization of the situation of the animals, pens and handling in this regard. What was the vaccination program? What was the routine of cleaning, disinfecting, emptying used?

L67 = there is no reference to the sex of the animals and the reference to the recommendation of the nutritional levels used, which is important considering that the nutrition of the animals can influence and be influenced by probiotics.

Table 1. L216 = what does “rosta” mean? The amounts of each ingredient are not in g / kg as it says. Why are the ingredients in g / kg and the chemical composition in%? What do the Title 1, Title 2, Title 3 columns mean?

L68 = “Fatteners from the experimental group…” All the animals used belong to the experiment, so if you want to refer to the animals of a given treatment, quote it.

Reverse the order of the second and third paragraphs in this section.

L74 = "Control and Experimental group ..." I do not think it is correct to name one group experimental and another not, after all, both groups were used in the experiment. I suggest naming it a control group and a probiotic group.

L79: 2.2. Performance traits

Has the feed supply been controlled? Why were the results of feed intake numerically the same?

If there was no ad libitum feed intake, it is necessary to highlight, as this totally changes the results of weight gain and feed conversion.

L86-87 = what do you mean by “fatteners from a insertion no. 86 IV (in the number of 60)… ”?

Results

Standard error values ​​of the mean and statistical probability in each variable must be presented in the tables.

Table 2 .: Was there an initial difference of approximately 5 kg in the animals' initial weight, was this item used as a covariate for the analysis of the other variables?

There is something wrong with the results, the final weight or the length of the period or the ADG of the Bioplus treatment does not match! HCW and dressing results also need to be reviewed.

Table 5 and 6: When standardizing the slaughter weight between treatments (approximately 112 kg), all carcass characteristics tend to match, canceling a possible effect of treatments on the variables!

Higher concentrations of B. subtilis and B. licheniformis were found in diets and in animals fed with probiotics, which was expected.

Discussion

This section is quite superficial in terms of discussing the results obtained. Arguments raised in the introduction could assist in this section.

L247-271 = there is a sequence of reports that are often disconnected from the results of the present study. Every framework of information from previous studies should be used in order to explain the present results, otherwise, it will be presented as a mere literature review.

Conclusions

The conclusions need to be concise and respond to the objective of the work. Short and direct phrases usually help to solve the problem to be investigated and presented at the beginning of the manuscript.

In this manuscript the conclusions are similar to a summary and are inaccurate.

Reviewer 3 Report

The Authors aimed to evaluate the effects of the dietary supplementation of a probiotic mixture (Bacillus licheniformis 16 DSM 5749 and Bacillus subtilis DSM 5750) in fattening pigs. , prior to slaughter and after a fasting period of 17 hours. Performance traits, carcass and meat quality, and composition of the gut microflora were investigated.

From a general point of view, the subject of the study is worth of investigation. However, there are several remarks that make the manuscript unsuitable for publication.

  1. The manuscript is poorly written and requires a careful English editing.
  2. At the beginning of the introduction (lines 31-36), the Authors omitted that the use of alternatives to antibiotics has been mainly prompted by the ban of antimicrobials as growth-promoters in many countries (e.g., since 2006 in EU).
  3. Lines 49-56: Authors stated that several studies investigated the effects of probiotics on pig health and growth performance, but data on carcass and meat quality are lacking. Thus, their main aim was to check such effects. However, during the study they focused their attention on gut microflora composition and performance traits.
  4. The number of animals used in the study is not clear. The Authors claimed that 87574 pigs were involved. However, the samplings for the evaluated parameters were performed on a very reduced number of animals.
  5. The sampling of gut content and gut mucosa is not properly described.